# User perspectives and preferences on a novel TB LAM diagnostic (Fujifilm SILVAMP TB LAM)–a qualitative study in Malawi and Zambia

Yannis Herrmann[1][☉]*, Federica Lainati[1][☉], María del Mar Castro[1], Chanda P. Mwamba[2], Moses Kumwenda[3], Monde Muyoyeta[4], Tobias Broger[1], Norbert Heinrich[5,6], Laura Olbrich[5,6], Elizabeth L. Corbett[7], Shannon A. McMahon[8,9], Nora Engel[10‡], Claudia M. Denkinger[1,11‡]*

1 Division of Clinical Infectious Disease and Tropical Medicine, Centre for Infectious Diseases, Heidelberg University Hospital, Heidelberg, Germany, 2 Centre for Infectious Disease Research in Zambia, Social & Behavioural Science Group, Lusaka, Zambia, 3 Malawi-Liverpool-Wellcome Clinical Research Programme (MLW), Public Health Group, Blantyre, Malawi, 4 Centre for Infectious Disease Research in Zambia, Tuberculosis Department, Lusaka, Zambia, 5 Division for Infectious Diseases, LMU Hospital, Munich, Germany, 6 German Centre for Infection Research (DZIF), Partner Site Munich, Munich, Germany, 7 London School of Hygiene and Tropical Medicine, Infectious and Tropical Diseases, London, United Kingdom, 8 Heidelberg University Hospital, Heidelberg Institute of Global Health, Heidelberg, Germany, 9 International Health Department, Johns Hopkins Bloomberg School of Public Health, Baltimore, MD, United States of America, 10 Maastricht University, Department of Health, Ethics & Society, Research School for Public Health and Primary Care, Maastricht, The Netherlands, 11 German Centre for Infection Research (DZIF), Partner Site Heidelberg University Hospital, Heidelberg, Germany

☉ These authors contributed equally to this work.
‡ NE and CMD also contributed equally to this work.
* claudia.denkinger@uni-heidelberg.de (CMD); y.herrmann@stud.uni-heidelberg.de (YH)

**Data Availability Statement:** The data set that support the findings of this study are openly available in the "heiDATA - Heidelberg Open

## Abstract

Widely available tuberculosis (TB) diagnostics use sputum samples. However, many patients, particularly children and patients living with HIV (PLHIV), struggle to provide sputum. Urine diagnostics are a promising approach to circumvent this challenge while delivering reliable and timely diagnosis. This qualitative study in two high TB/HIV burden countries assesses values and preferences of end-users, along with potential barriers for the implementation of the novel Fujifilm SILVAMP TB-LAM (FujiLAM, Fujifilm, Japan) urine test. Between September 2020 and March 2021, we conducted 42 semi-structured interviews with patients, health care providers (HCPs) and decision makers (DMs) (e.g., in national TB programs) in Malawi and Zambia. Interviews were transcribed verbatim and analyzed using a framework approach supported by NVIVO. Findings aligned with the pre-existing Health Equity Implementation Framework, which guided the presentation of results. The ease and convenience of urine-based testing was described as empowering among patients and HCPs who lamented the difficulty of sputum collection, however HCPs expressed concerns that a shift in agency to the patient may affect clinic workflows (e.g., due to less control over collection). Implementation facilitators, such as shorter turnaround times, were welcomed by operators and patients alike. The decentralization of diagnostics was considered possible with FujiLAM by HCPs and DMs due to low infrastructure requirements. Finally, our findings

Research Data" repository at https://doi.org/10.11588/data/PLRQCZ.

**Funding:** This study was supported by the German Center for Infection Research [TTU.02.909; https://www.dzif.de/en]. The grant recipient was CMD. For the publication fee we acknowledge financial support by Deutsche Forschungsgemeinschaft within the funding programme "Open Access Publication Costs" as well as by Heidelberg University. The funder had no role in study design, data collection and analysis, decision to publish, or preparation of the manuscript.

**Competing interests:** I have read the journal's policy and the authors of this manuscript have the following competing interests: CMD was working for FIND until April 2019. FIND at the time partnered with Fujifilm to develop the the SILVAMP TB Assay.

support efforts for eliminating the CD4 count as an eligibility criterion for LAM testing, to facilitate implementation and benefit a wider range of patients. Our study identified barriers and facilitators relevant to scale-up of urine LAM tests in Malawi and Zambia. FujiLAM could positively impact health equity, as it would particularly benefit patient groups currently underserved by existing TB diagnostics. Participants view the approach as a viable, acceptable, and likely sustainable option in low- and middle-income countries, though adaptations may be required to current health care processes for deployment.

**Trial registration:** German Clinical Trials Register, DRKS00021003. URL: https://www.drks.de/drks_web/setLocale_EN.do

## Introduction

Tuberculosis (TB) remains the leading cause of death among people with human immunodeficiency virus worldwide [1]. In sub-Saharan Africa, TB accounts for approximately 40% of hospital deaths among people living with HIV (PLHIV), of which more than half are undiagnosed before death [2, 3]. The current diagnostic standard in countries with high TB burden, including Zambia and Malawi, focuses on sputum-based tests which have limitations in diagnosing TB in particularly vulnerable groups such as PLHIV and children. The recommended sputum-based TB test in Zambia and Malawi, Xpert MTB/RIF and Xpert Ultra MTB/RIF (Cepheid, Sunnyvale, CA, USA; henceforth called Xpert), is widely used for TB diagnostics across different countries [4]. PLHIV, especially those who are severely ill, and children often cannot produce sputum, which makes a sputum-based diagnosis challenging. When compounded with the low sensitivity of smear microscopy, as well as a limited access to molecular diagnostics, patients often lack a confirmed diagnosis [5, 6]. Moreover, patients suffering from extrapulmonary TB, are missed by sputum-based testing methods [7, 8].

Urine-based assays that can detect TB-related antigens, namely lipoarabinomannan (LAM), are a promising approach to overcome sputum-based TB testing challenges. The first-generation urine-based TB LAM test, Abbott's Alere Determine TB-LAM (USA; henceforth called AlereLAM), has shown to reduce mortality from TB disease in patients with advanced HIV, despite a limited sensitivity, due to timely results and earlier initiation of treatment [9, 10]. A next-generation TB test, the Fujifilm SILVAMP TB-LAM assay (Japan; henceforth called FujiLAM), also uses urine and offers higher sensitivity compared with AlereLAM for TB detection in adults with HIV across CD4 count strata, as shown in studies on biobanked samples [5, 11–13] and in a first prospective study using fresh urine [14]. There is potential value of FujiLAM in an HIV negative population and potentially also in children [15, 16]. A recent modelling analysis estimated that 120,000 deaths and 420,000 incident TB cases could be averted between 2020 and 2035 by implementing FujiLAM [17]. Both FujiLAM and Alere-LAM tests are rapid and usable without additional instrumentation, however, FujiLAM involves additional steps (including timed steps), takes longer to perform (50 vs 25 minutes), and might be more costly (estimated $6 vs $3) [5, 18]. Currently, the use of AlereLAM is recommended for severely ill patients and immunocompromised patients with low CD4 counts according to the World Health Organization (WHO) [19]. This requires AlereLAM to be used in conjunction with blood-based tests to determine the patient's CD4 count. These aspects do affect feasibility, usability and acceptance which further affects AlereLAM's implementation [20]. A recent survey has shown that only 30% of countries have incorporated LAM-based testing in their policies [21]. FujiLAM is not yet commercially available and is currently evaluated

for its accuracy in adults and children as part of two multi-center studies led by FIND (NCT04089423) and the RaPaed TB consortium (NCT03734172).

Failing to consider the perspective of stakeholders, such as patients, healthcare providers (HCPs), and decision makers (DMs), in the introduction and scale-up of new diagnostics often results in slow scale-up, inappropriate and limited use, and consequently limited added value [20, 22, 23]. In this multicenter qualitative study in Zambia and Malawi in public hospital settings, we explore perspectives on usability and implementation of FujiLAM, for adults and children (and their guardians) presenting for TB care in the context of a diagnostics accuracy study on FujiLAM. Findings are presented following the health equity implementation framework [24], with barriers and facilitators described at different levels including the patient, provider, patient-provider interaction (or the clinical encounter), characteristics of the innovation, and the healthcare system (inner and outer context). The framework identifies factors that promote or impede implementation of novel treatments while assessing health equity aspects.

## Methods

### Study design, data collection and participants

We conducted 42 semi-structured interviews with participants from Malawi and Zambia including adult patients living with HIV (PLHIV); guardians of children with presumed or confirmed TB; healthcare providers (HCP); and national and local decision makers (DM) in TB programs (Table 1). HCPs included nurses, research assistants, clinicians, and lab technicians. DMs were individuals who held a direct role in the development of national and regional guidelines, and/or those who made operational and administrative decisions in TB programs (ministry officials, advisors to ministries, and physicians in senior roles in clinical and nonprofit organizations). In addition, one focus group discussion was held in Zambia with five health care providers and one lab technician. The study took place in two urban public

**Table 1. Participants overview per country.**

| Country | Type of Participant | No. of interviewed Participants |
|---|---|---|
| **ZAMBIA** | | |
| **Public Urban Hospital Setting** | Health Care Providers (HCP) | |
| | Nurses | 1 |
| | Research Assistants | 2 |
| | Clinicians | 2 |
| | Lab Technicians | 2 |
| | Decision Makers (DM) | 3 |
| | Patients with presumed TB (P) | 11 |
| **MALAWI** | | |
| **Public Urban Hospital Setting** | Health Care Providers (HCP) | |
| | Nurses | 4 |
| | Research Assistants | 2 |
| | Clinicians | 1 |
| | Lab Technicians | 3 |
| | Decision Makers (DM) | 2 |
| | Patients with presumed TB (P) | 4 |
| | Guardians of children with presumed TB (G) | 5 |
| | | 42 |

hospitals in Malawi and Zambia that provide first line TB and HIV treatment and where, at the time, patients were enrolled in the FIND diagnostic accuracy study of FujiLAM (NCT04089423, Clinicaltrials.gov), and the RaPaed–TB diagnostic accuracy study for use in children (NCT03734172, Clinicaltrials.gov). In order to place our study into context, a description of the health care settings and current Urine-LAM diagnostic algorithms in Malawi and Zambia is presented in the S1 File.

HCPs, LTs and DMs were purposively sampled from the participants of the prospective accuracy studies to reflect the various professional cadres. As FujiLAM is not yet commercially available, no participant had experience with the test outside of a study setting. Patients/guardians were enrolled consecutively based on the availability of prospective TB accuracy study interview staff. Patients, including minors, were either presumed or recently confirmed to have TB and thus, under active investigation for TB, or had recently completed a diagnostic process. Professionals (HCP, LT, DM) involved in the accuracy study were contacted and invited for an interview. All interviewees or their guardians provided written informed consent.

Interviews with patients, guardians, healthcare workers and lab technicians were conducted face-to-face at the clinics by the local investigators (CPM, EH, LC, MK, HS) following semi-structured interview guides for each user group as follows: for patients and guardians, topics included background about the patient, diagnostic processes for HIV and TB, sample collection, time-to-results, accessing facilities for testing and follow-ups. For HCPs, topics included their knowledge on the current TB diagnostic landscape for PLHIV and children, sample, testing capacity, results, follow-up, diagnostic yield, and specific questions on the FujiLAM kit. Interviews with DMs were held by teleconference (YH) using a semi-structured interview guide specifically for this group. For DMs, topics included the current TB diagnostic landscape, their knowledge on FujiLAM, their views on its feasibility of implementation, and its potential impact on health equity. Interviews were conducted in English or the local language and lasted 30 to 90 minutes. After informed consent, all 42 interviews were audio-recorded and complemented with written notes taken by the local investigator.

## Data entry and analysis

Audio files were transcribed and, when necessary, translated into English. YH and FL checked the files for accuracy and requested edits or clarifications as needed. Once data saturation was reached, data collection concluded and the team followed a 5-step analysis approach as outlined by Pope et al. [25]. YH and FL each immersed themselves in the data by reading and re-reading transcripts (Step 1), and then began discussing codes (Step 2). Some codes were deductive, drawn from the interview guides and broader research aims, but as line-by-line coding commenced, coding was also inductive: as new ideas emerged, they were added to the codebook and applied across the full dataset (the codebook is included as a supplement (see S2 File). YH and FL used NVivo 13 to support coding. Subsequently, the coded material was organized into themes using thematic analysis (Step 4) [26–28].

The analysis was conducted by YH, FL and MCN, and closely supervised by NE and CMD. This fuller team discussed the themes and sought to identify existing theories or frameworks that could guide presentation of the data (Step 5), ultimately identifying the Health Equity Implementation Framework [24]. The HEIF draws together conceptual guidance from implementation science and health disparities literature. Aligning with the HEIF, data were divided into barriers and facilitators, including those that are unique to vulnerable populations [24, 29], such as children and PLHIV. We consider facilitators as factors that enable the implementation of evidence-based interventions [30]. Also, data were arranged according to

determinants of implementation [31] at the levels of: (1) context (e.g. system-level mandates that might overwhelm staff, or a clinic culture's openness to changing practices); (2) recipients (e.g. patients and guardians); and (3) characteristics of the innovation (e.g. regarding the operation of FujiLAM). Inner factors at the local level can include leadership support for an innovation or feedback processes; while outer context factors relate to the wider health system, policy, social, regulatory and political infrastructures surrounding the local context [31]. Findings related to the clinical encounter or the interaction between recipients (e.g., patient and provider) and the innovation itself (e.g., test), proved especially salient in our study and are detailed in the results.

The data that supports our findings is accessible through the heiDATA network of the University of Heidelberg [32].

## Inclusivity in global research

Additional information regarding the ethical, cultural, and scientific considerations specific to inclusivity in global research is included in the S3 File.

## FujiLAM test characteristics and use instructions

FujiLAM is currently being evaluated in prospective diagnostic accuracy studies in adults (NCT04089423, clinicaltrials.gov) and paediatric patients (NCT03734172, Clinicaltrials.gov). FujiLAM's is an instrument-free platform; the test procedure is illustrated in a video and takes 50–60 minutes from start to end (https://www.youtube.com/watch?v=aK-QtzkLBug). In brief, urine is added to the reagent tube up to the indicator line, mixed, and incubated for 40 minutes. Then 2 drops are added to the sample port, and immediately after, button 2 is pressed. After 3 to 10 minutes, the "go-next" color indicator mark on the reading window turns orange, and button 3 must be pressed. Within the next 10 minutes, the result is read. The presence of the control line only is interpreted as a negative result; the presence of the control line plus the test line is interpreted as a positive result; the absence of the control line is interpreted as invalid.

## Ethics

The study was approved by the Ethics Committee of the Heidelberg University Hospital, approval number S-003/2020; the University of Malawi College of Medicine Research and Ethics Committee, approval number P.01/19/2569, the University of Zambia Biomedical Research Ethics Committee, REF. No. 486–2019, and the National Health Research Authority of Zambia (Lusaka, Zambia). Written informed consent was obtained prior to participation.

## Results

We present the perspectives of 42 patients, healthcare providers (HCPs) including lab technicians (LTs), and decision makers (DMs) regarding FujiLAM implementation (Table 1). The themes are presented within the different levels of the health equity implementation framework and summarized in Table 2 [24]: a) innovation factors, b) clinical encounter, c) recipient factors (patient and providers) and d) context.

### Innovation factors

**FujiLAM is simple to use but contains too many (time-sensitive) steps for use as a POC test.** Professionals who have operated the test consider it simple, and the steps are easy to follow using manufacturer's instructions. Therefore, while training is necessary prior to operating

**Table 2. Barriers and facilitators to the implementation of FujiLAM, described by innovation factors, patient-provider interaction (or the clinical encounter), recipient factors (patient and provider), and the context factors (inner and outer).**

| | | Barriers | Facilitators |
|---|---|---|---|
| Innovation factors | | The multiple timed steps and increased hands-on time, which may affect the ability to process in batches (relevant in facilities with high volume of patients) and for point-of-care testing | No need for specific expertise, just a quick training<br>Result interpretation is considered easy by LTs, and does not require visual interpretation of test-line intensity as opposed to AlereLAM, thus perceived as less subjective.<br>Low maintenance and electricity requirements, since no additional equipment/machine is needed<br>1-hour TAT may allow same-encounter diagnosis<br>Increased sensitivity and high positive predictive value (PPV) to rule-in TB compared to other LAM-based tests and good diagnostic yield |
| Clinical encounter | | The potential shift in agency from the HCPs to the patients during outpatient sample collection (urine for LAM testing) | Good communication and the feeling of being taken cared for may be a motivating factor to return to the clinic |
| Recipient factors | Patients | Patients need to travel long distances to access TB diagnostics and treatment | Urine is more convenient to produce, compared to sputum<br>Do not need to face the stigma/ discomfort around sputum samples<br>Privacy of toilets for urine collection is appreciated |
| | Providers | Perceived risk of losses-to-follow-up (e.g., patient do not return with the sample)<br>Need for scientific publications to gain trust in FujiLAM | Appreciation of urine as an easy sample for patients to produce |
| Context factors (inner and outer context) | | Challenges to same day diagnosis (processes at the lab, staff shortages or limited staff to process the tests) might not allow point-of-care diagnosis and treatment initiation during the first encounter, even with a 1-hour TAT<br>From the lessons learned from AlereLAM: issues such as limited availability and difficulties maintaining the supply of the test at the POC are potential challenges to the deployment of FujiLAM<br>Limited availability of functioning and private sanitary facilities to collect samples<br>Criteria for eligibility to use LAM testing (requirement of CD4 count like in AlereLAM WHO guideline) is perceived as being too restrictive | Program directors and other professionals perceive that implementation could be possible at primary care facilities<br>No need for laboratory infrastructure |

the test, no specific laboratory expertise is needed. In addition, HCPs consider that FujiLAM is easy to interpret, particularly when compared to AlereLAM. Interpretation of AlereLAM requires comparison of the result line with a reference scale card, which leads to difficulties interpreting faint lines, or affecting the quality of results if the reference card is not applied properly. A DM highlights this as a main barrier of AlereLAM compared to FujiLAM:

> "I think the FujiLAM is easier to interpret than the Alere because the Alere has a lot of subjectivity on how you see line, whether it is faint or not. This is not a common issue with Fuji [LAM]" (DM, Malawi).

Troubleshooting with FujiLAM is also considered straightforward, for example, when the control line is not present, users know they need to re-do the test. Due to the simplicity of the test operation and result interpretation, a wide range of HCPs should be able to carry out the test, for example nurses, independent of laboratory expertise.

However, some users indicate that there are too many timed steps, and the test requires too much hands-on time. Consequently, professionals suggest that FujiLAM should be operated by technicians in the lab, not by nurses or clinicians (e.g., at the bedside) as they hold other roles which could be interrupted while waiting for the timed steps. The number of steps may also affect the capacity of running multiple tests in parallel (batches) at different times and therefore limit throughput, particularly in facilities with staffing limitations. These issues are highlighted by a DM in Malawi:

*"(. . .) the only limitation is the timing issues that are associated with Fuji[LAM]. If you are at a high-volume primary health care [facility] where you have so many samples and maybe you have to run some in bulk [at] different times, it could be complicated (. . .)."* (DM, Malawi)

**Low infrastructural and maintenance requirements for test operation.** The FujiLAM kit allows running the test without relying on much additional equipment or maintenance (besides personal protective equipment, urine cup, etc.). This is contrasted with Xpert's infrastructure requirements, whose machinery and modules need installation, electricity, and maintenance, and must therefore be handled by laboratory technicians who received extensive training to properly manage the devices. For these reasons, professionals consider that Fuji-LAM is suitable to use in remote areas and decentralized laboratories in Malawi and Zambia. Some of them recommend that implementation should be prioritized in facilities that do not have technologies such as Xpert and microscopy, to expand the availability of TB diagnostics in remote areas. Patients highlight the importance of wide-spread implantation and a DM summarizes some enablers for FujiLAM's implementation:

*"It has to be introduced in all the facilities whether small or big."* (P, Zambia)

*"This test method should be accessible in different areas such as in the villages so that people shouldn't be travelling far."* (P, Malawi)

*"(. . .) it [FujiLAM] doesn't require any supportive equipment; it doesn't need anything beyond the fact that you need a room. You don't need any machine, or any expertise. (. . .) it's a fairly easy step-by-step process that once you teach someone, they should be able to do it with no problems"* (DM, Malawi)

However, a potential issue for this is the kit's storage temperature of 1–35˚C which may be difficult to keep in certain areas of Malawi and Zambia.

**FujiLAM's increased sensitivity and diagnostic yield.** The higher diagnostic sensitivity of the next-generation LAM test drives the desire of professionals to use FujiLAM over Alere-LAM to correctly identify more TB cases. AlereLAM's low sensitivity is reported as one of the reasons for the limited uptake and slow implementation. Patients also appreciate a higher sensitivity as it gives them more comfort that they have been correctly diagnosed. Therefore, Fuji-LAM's higher sensitivity is seen as an important factor when evaluating its possible implementation and is particularly appreciated for hard-to-diagnose patients such as PLHIV and children:

*"I like that FujiLAM is better than AlereLAM because its increased sensitivity is ideal for children (. . .) and so at least we have something in our corner which is a little bit better in terms of performance in children, and also in HIV-positive patients. I think that in our setting it definitely helps to fulfil that need that is there."* (DM, Malawi)

In contrast, some DMs consider that a sensitivity of around 60% is still low, and that a higher sensitivity (at least 70%) should be reached to justify the opportunity cost of a rollout. It is, however, acknowledged that an imperfect sensitivity can be compensated by combining FujiLAM with other tests or relying on clinical diagnosis. A DM explains the importance of a high sensitivity:

*"We need a test that is highly sensitive. So, results around 50–60% wouldn't actually change anything. If a test has a low sensitivity, it becomes so unreliable. (. . .) If a test misses a huge*

*number, it can be costly to rule it [TB] out and it will misguide the decision making. (. . .) I think improvement to the FujiLAM is still needed because the sensitivity is higher than the Alere one but still not where we would like it to be"* (DM, Zambia)

**Quick turnaround time (TAT).** Participants appreciate the possibility of a same-encounter diagnosis with FujiLAM, given its TAT of less than 1 hour. The importance of prompt diagnostic results is underlined by all participants for different reasons. For example, HCPs and patients likewise state that it would decrease the patient's stress while waiting and worrying about the potential outcome of the test, increase patient satisfaction, reduce costs to the patients for multiple trips to the health care facilities, and reduce loss to follow-up and failure to initiate treatment, as is the case with currently available diagnostic tests. Clinicians and DMs suggest that same-day results to inform decision making will potentially decrease the movement of potentially infectious patients and therefore the risk of spreading TB, while providing earlier initiation of treatment. Also, same day diagnosis will likely increase the number of people effectively starting treatment and reduce loss to follow up. Interviewees considered that the test would fit in their daily routine (urban, public hospital), due to its fast TAT which allows to quickly confirm suspected TB, including a study clinician in Zambia:

*"It [FujiLAM] can, easily [fit in current daily routine]. And it would make work so much easier because the ones that are highly suspicious, you knew that the test is probably going to come out positive, you don't have to wait 24 hours to make a diagnosis. (. . .) So at least it can help you make a decision right within hours of seeing the patient without having them to come back the following day."* (HCP, Zambia)

## Clinical encounter

**A shift in agency from the HCPs to the patients during outpatient sample collection.** The collection of urine, as opposed to sputum or blood samples, is described as "easy" "convenient" and "unproblematic" across groups. However, a potential barrier to implementation identified by professionals is the shift of agency from the HCP to the patient during the process of sample collection. In outpatient settings, the LAM sampling process relies on the patient's ability to produce the urine sample. However, patients, especially children, may not be able to urinate on the spot. There is a perceived risk of loss to follow-up or delay in diagnosis in situations when the patient needs to go home to collect the sample and an overall feeling of loss of control by the HCP. In contrast, a blood samples (which may be taken for HIV testing) gives more agency to the HCPs, as the sample can be collected from the patient whenever it suits the HCP's workflow. The same applies to inpatient urine collection in catheterized patients.

*"You can collect it [urine sample] from anybody, even from a patient who is unconscious."* (DM, Zambia)

**Good communication and the feeling of being taken cared for may be a motivating factor to return to the clinic.** Good communication between the patient and the HCP is a facilitator in this process, since a patient's feeling of being taken cared for may be a motivating factor to return to the clinic for the results and continue follow-up. Some patients consider the time of receiving the results as secondary and prioritize a reliable diagnosis through the right test and proper sample processing. This variability in how much patients value the TAT seems to be independent of the distance they must travel to reach the testing facilities or how sick they feel, but rather their perception that healthcare workers are doing

their best to achieve an accurate diagnosis. Two patients point out why the time-to-result may be of secondary importance:

> *"I didn't get the results on Friday and today they didn't give me the results as well. (. . .) I don't see it as a problem because they have to test the samples properly"* (P, Malawi)

> "*it's good that they test and start checking up on you, we are processing your results, maybe tomorrow or the next day, we will let you know so that will make me see that these people care about me*" (P, Zambia)

### Recipient factors: Patients

**Urine is more convenient to produce than sputum, but privacy for sample collection is important.** One of the main facilitators for FujiLAM's implementation is the use of urine as a sample for TB diagnosis. Sputum samples are known to be difficult to collect, specifically in children and severely ill adult patients. Patients primarily express discomfort or difficulty to produce the sputum sample. Sputum induction or invasive procedures to support sputum collection such as gastric aspiration, commonly used in children, are often uncomfortable and painful for the patient. Therefore, participants see urine-based collection, especially also among children and adults who cannot produce sputum, as advantageous.

> *"TB is like a silent killer for people living with HIV, you can't know what is eating you up but you will find that you are just getting sick because TB comes with different symptoms. (. . .) and I was happy that they are now using urine also. I saw that things are changing because I used to be one of those people who fail to produce sputum that if they fail to produce they can check in urine (. . .). Urinating is not difficult you just have to drink a lot of water that's all . . .now coughing you can't even force yourself to cough (. . .)"* (P, Zambia)

In terms of physical infrastructure to facilitate collection, patients state that privacy is very important to them, and appreciate the privacy of bathrooms to collect urine. For sputum collection there is usually no designated area to provide the privacy patients wish for. This is particularly problematic when considering the stigmatization of sputum as a sample in many high TB burden settings. Nonetheless, privacy challenges inherent to urine collection occur when sanitary facilities are not functioning, are unhygienic, or are considered unsafe.

**Patients may need to travel long distances to access TB care.** Many patients seem to value same-encounter diagnoses because of the inconveniences to reach healthcare facilities that provide TB care, which is an important consideration when evaluating feasibility of decentralizing FujiLAM. For example, most patients must reach the hospital on foot, and in some cases, the distances having to be travelled are extensive. This seems to especially be a problem in patients who are very unwell from TB and suffer from severe fatigue, who discuss the inability to walk. While TB testing is free, most patients outside a study setting will have to cover the cost for transportation to and from facilities. Some patients and guardians complained about not having adequate transport money to travel health facilities. Other study participants who could afford to pay their transportation costs to a health facility describe the journey as sometimes long and having to take multiple minibuses. Additionally, limited transportation infrastructure between rural and central health facilities (such as district and central hospitals) is described, and patients need to look for alternative facilities, forms of transport or simply ignore their clinic appointments, leading to undetected cases.

TB diagnostics strategies for special patient groups, considered hard-to-diagnose, such as children or patients suffering from extrapulmonary TB, are not available in many clinics in Malawi. Several interviewed guardians reported having to approach three different facilities before their child could receive proper screening and care.

### Recipient factors: Providers

**Appreciation of urine as an easy sample for patients to produce.** HCPs acknowledge that urine is easier to collect than sputum for several patient groups. Invasive procedures to support sputum collection, such as gastric aspiration, not only are challenging to the patients but require trained personnel and a license, and relies of additional protective equipment. Therefore, participants see urine-based collection as advantageous, especially among certain population groups who find sputum collection more difficult.

**Need for scientific publications to gain trust in FujiLAM.** DMs point to the importance of increased awareness, experience and scientific publications on a test that will guide a clinician's trust for the test used and understanding of FujiLAM. This, in turn, also ensures an interest in and demand for the test:

*"[There] is a self-created demand and more publications are highly required."* (DM, Zambia)

### Context factors

**Criteria for CD4 count and its potential impact on FujiLAM's use and implementation.** According to current guidelines, AlereLAM testing is only recommended for a specific subset of the population of HIV positive patients (defined by CD4 count), which is perceived as a barrier for implementation by health programs. Requirements of an HIV diagnosis and a CD4 count for eligibility for FujiLAM testing, would similarly be a limiting factor for its implementation (DMs). DMs express the wish to drop the CD4 count to determine eligibility of LAM tests, also based on the consideration that progress in HIV care and current treatment regimens for PLHIV has led to higher CD4 counts. Furthermore, for pediatric patients, DMs want a urine test that could be applied regardless of HIV status, due to the difficulty of sputum production and low proportion of pediatric TB that is currently confirmed. A DM talks about the drawbacks of having a CD4 count restriction:

*"It [AlereLAM] is only applied at a particular population and within that population it is limited to a population with a low CD4 count. So it becomes quite challenging for programs to implement it with all these nitty-gritties around it. (. . .) and the current process in HIV is shifting away from CD4 by a lot. That process in CD4 creates a bias because not every patient's CD4 is low. So we need more studies that look more at the clinical definition of who can benefit from the urine LAM. (. . .) If that restriction on CD4 can just go away, I think that would be most useful."* (DM, Zambia)

Some DMs and clinicians express the concern that people might start using the test excessively without following the guidelines, thus applying the test in a patient group that is not eligible, e.g. patients whose CD4 count does not meet the criteria, and this would result in potentially false conclusions from the results. For this reason, DMs in particular highlight the need for raised awareness and training to increased understanding of the benefits and limitations of this novel test.

As an alternative to overcome this barrier, a POC test for determining the CD4 count (Omega's Visitect CD4 test [33, 34], henceforth called Visitect) was explored. HCPs agree that the Visitect test can be implemented alongside LAM-tests, since these tests are easy to conduct. HCPs in Zambia indicate that the fast result of the finger prick test would be of value to integrate into routine diagnostics of hospital visits to have a timely idea of the patient's immune status. This, considering that CD4 count testing is usually not available in a routine setting on the day of admittance.

*"I would recommend it [Visitect] especially in the newly diagnosed patients so that it can be done on the day of enrolment so that we have an idea about the immunity of a patient."* (HCP, Zambia)

**Regardless of turnaround time, same-day diagnoses remain difficult to achieve.** As previously described, LTs and HCPs value the short turnaround time of FujiLAM. However, even with a quick TAT, same-day diagnosis is difficult to realize. Dynamics related to the processing of samples and workflow at the facilities which also affect the delivery of the TB test results to the patients, may limit a same-day diagnosis irrespective of TAT for a specific test. At some healthcare facilities, only one lab technician might be available, who then would have to work through all the samples. This limits the timely processing and reporting of the tests. Additionally, lab technicians may prefer to test in batches to keep track of samples more easily, which means that the samples are kept on hold until enough samples are collected to be processed together, resulting in diagnostic delays. A clinician explains why it would be difficult releasing the results the same day, even with a short TAT, by recounting the experiences with Xpert:

*"So those ones like I said, the turnaround time is 24hrs and the test itself is done in 2hrs, but because there are a lot of samples that they work on and they have to report them, it takes a bit longer (. . .) this is a facility that has probably like 70 samples that need to be run in a day so, (. . .) they usually just ask us to give them a 24hr turnaround time to make it easier to work properly and report."* (HCP, Zambia)

HCPs and DMs consider that FujiLAM will not change the time to results if the issues in the process of testing and releasing the results on the same day continue. This considering that FujiLAM only saves one hour compared to Xpert, again indicating that the processes and not the tests themselves currently prohibit a same day result release.

*"Well, I think it [FujiLAM] does help them to access treatment at an earlier stage and I think it also depends on the effectiveness of the process that is offered at the facility. (. . .). So, if there is anything that is, you know, hindering that process ensuring that the patients get the result the same day before they go home then it [FujiLAM] doesn't really improve anything (. . .)."* (DM, Malawi)

**Generating the demand and meeting supply.** Participants highlight a need to generate demand and meet the supply. In Malawi, a decision maker states that the availability of TB diagnostics is depended on its allocation by the government, and therefore if FujiLAM is not prioritized at the level of the ministry, health facilities will not pick it up. Furthermore, the demand for LAM tests needs to be increased by clinicians, which will go together with their increased knowledge of the test through increased communication and scientific publications on FujiLAM. HCPs have also noticed some initial reservations on the patient's side due to the

unfamiliarity of urine as a sample to detect TB, thus generating awareness among patients who come for TB testing is also important when implementing LAM testing.

Consequently, once the demand is generated, having an adequate supply chain to serve all levels of the health care system is critical to meet needs. Due to apparent delivery issues of AlereLAM, the test is currently only available in small quantities across Zambia. In Malawi, a similar situation is reported by a head clinician who says that even though AlereLAM is part of the Malawian Tuberculosis guidelines, it is not routinely available at sites.

## Discussion

The results of our study show that overall, the Fujifilm SILVAMP TB LAM test was appreciated by patients, providers, and implementers and could have a positive impact for patients with presumed TB, particularly for patient groups currently underserved by existing TB diagnostics, such as children and PLHIV. Facilitators for implementation include its ease of use and clear interpretation, short turnaround time, the convenience of urine as a sample, and low infrastructure requirement. FujiLAM was perceived as suitable for use at decentralized facilities in Malawi and Zambia, with the potential to improve access and equity. The higher sensitivity and diagnostic yield drive the preference of this test over AlereLAM for implementers. However, some barriers such as the number of the time-sensitive steps, increased hands-on time and the potential restriction on eligibility based on CD4 count were also identified. We used the health equity implementation framework to structure our results [24]. We present the barriers and facilitators with a focus on potential healthcare disparities, and factors that are relevant to these vulnerable groups, to help guide implementation of FujiLAM.

From the recipient (patient) perspective, urine-based LAM testing is generally seen a promising approach to TB diagnosis, as it reduces the reliance on sputum, a specimen challenging to provide for many patients [5, 35]. Patients and caregivers describe urine as convenient to produce compared to sputum. Furthermore, urine has the advantage of being a less stigmatized sample. These findings are supported by previous reports on AlereLAM [36] and a recent analysis showing that a higher proportion of PLHIV were able to produce urine compared to sputum for TB diagnosis [37]. Introducing a sample that is easier to produce for hard-to-diagnose patient groups will have a positive impact on health equity. However, the benefits of urine as a sample are limited by the availability of sanitary facilities that are hygienic, safe and ensure privacy. This is relevant for small or rural clinics, since the lack of sanitary facilities may impede urine collection [38], causing delays in diagnosis and risking loss to follow-up. This potential loss to follow-up was highlighted by the HCPs, particularly in outpatient settings and patients such as children, who may not produce the urine sample on command. This aspect of the patient-provider interaction reveals a shift of agency from the HCPs to the patients during outpatient urine collection. HCPs value having control over the sample collection process and are concerned over how this affects workflow in the clinical setting. The shifts in agency between patients and HCPs should be considered during implementation and when balancing the needs of different users.

The potential need of a CD4 count cutoff to determine eligibility for LAM testing, was regarded as a barrier for implementation. This observation was based on current restrictions of the use of AlereLAM only in PLHIV with a low CD4 count [39]. The additional TAT for CD4 testing hinders the use of LAM as a bedside test and may lead to delays or the loss of patients if they are referred for CD4 testing [40]. Alternative diagnostics to determine CD4 count at POC, such as Omega's novel Visitect CD4, were used in the study settings as a promising approach to enable more rapid decision-making at POC [41]. In this context, it is also important to state that LAM-positive results are often confirmed with Xpert. This is of

particularly importance in high burden multi-drug resistant TB countries, such as Zambia, as Xpert can simultaneously determine rifampicin resistance [39], while LAM tests do not support resistance testing. As previously reported for AlereLAM [36], HCPs and DMs fear that restricting the population eligible for LAM testing would limit adoption of FujiLAM if the eligible population is small. Furthermore, CD4 count is not performed in some settings to avoid diagnostic delays [40] and limit costs [42]. If the recent evidence suggesting a FujiLAM sensitivity in the range of 53 to 75% in HIV-negative patients from endemic TB regions can be confirmed, then a recommendation independent of CD4 count and potentially independent of HIV status might be possible [14, 15]. If not, a careful consideration of optimal operationalization in the settings of intended use is critical to maximize the test's impact. This should include consistence and clear communication of testing algorithms and guidelines accompanied with training of healthcare staff.

FujiLAM's TAT of under 1 hour is an innovation factor appreciated by the operators and decision makers since it theoretically allows delivering the results to the patient on the same day or same clinical encounter. POC tests are intended to give a reliable result in a short period of time which then enables clinicians to provide timely care for the patients [43]. However, HCPs and DMs highlighted that same-day treatment initiation was still unlikely unless contextual factors including workflow and patient flows are addressed. In particular, adjustments to staffing at the lab and in facilities would be needed to decrease the TAT and avoid overburdening existing systems that already struggle with the workload [44]. Also, the timed steps of FujiLAM may pose difficulty to integrating it at the bedside and limit batch testing, particularly when compared to AlereLAM. The measures of workforce strengthening have been shown necessary for implementation of other tests [40, 45, 46]. Patients would much appreciate a same-day turn around rather than returning again as Rucker et al. have shown as well [40]. This would also reduce the risk of losing patients to follow-up [47]. FujiLAM may contribute to decentralization of TB testing given its lower training and infrastructure requirements. Decentralization of testing in HIV and pediatric care has shown to positively impact mortality and linkage to treatment while reducing the risk of losses to follow up, thus increasing health equity [48, 49].

This multi-site study provides perspectives of test users, patients, and decision makers to inform FujiLAM implementation in two high burden TB countries. The use of semi-structured interview guides, tailored to each group allowed for new topics to arise in a total interview time of >30 hours is a strength of our study. We adapted to the restrictions of the COVID-19 pandemic by holding five interviews using an online meeting platform, which has been used successfully in previous qualitative studies [50]. Analysis was conducted in English. A possible limitation might be the potential loss of relevant information during translation. Another limitation stems from the study setting. Interviewees were using FujiLAM in urban public hospitals, with many of the operators having previous experience with conducting research. Furthermore, urban hospitals may be better equipped in comparison to more decentralized primary health care facilities where patients often present first for care. Thus, the transferability of our findings to primary health care facilities might be limited as it relied on the assessment of participants presenting to or working at urban hospitals. In addition, the volume of patients was high. Therefore, some aspects, e.g., batch testing, may be different from what would be expected in decentralized, rural settings.

## Conclusion

Our study identified several barriers and facilitators for the implementation of Urine LAM tests in Malawi and Zambia. Urine-based testing for TB diagnosis bolstered a sense of ease and

convenience, particularly for groups facing additional vulnerabilities (PLHIV, children, stigma). However, these benefits are crucially dependent on the availability of hygienic, safe and private sanitary facilities. Shifts in agency over sample collection process can affect workflows and concerns over potential loss to follow up. Implementation factors, such as turnaround time were welcomed by operators and patients alike. An aspect considered particularly relevant to vulnerable populations to increase access, is the decentralization of the diagnostic process. This is possible with FujiLAM due to the low infrastructure requirements. However, investments in staffing and sanitary facilities are required. Finally, our findings support efforts for eliminating the CD4 count as eligibility criterion in patients living with HIV, to facilitate implementation and benefit a wider range of patients.

## Supporting information

**S1 File. Context: Current urine LAM guidelines and TB diagnosis algorithms in Malawi and Zambia in their respective healthcare settings.** A more detailed description of the current TB diagnostic algorithms in Malawi and Zambia is provided. In addition, we described clinical settings in both countries to allow a better understanding of the respective health care facilities where our study took place.
(PDF)

**S2 File. Codebook.** Final codebook developed after saturation of interviews was reached considering the interview transcripts, topic guides and field notes. This codebook was applied to the entire data set before establishing descriptive memos and themes.
(PDF)

**S3 File. Questionnaire on inclusivity in global research.** Our study team conducted research in collaboration with partners at study sites in Malawi and Zambia. Therefore, we added an additional questionnaire outlining ethical, cultural, and scientific considerations specific to inclusivity in global research for full transparency.
(PDF)

## Acknowledgments

We would like to thank all our participants for their valuable time and insights. Additionally, we want to thank Henry Sambakunsi, Esther Hamweemba and Lloyd Chifunda for helping in the conduct of the interviews at the study sites in Malawi and Zambia. The project was supported by FIND, the Global Alliance for Diagnostics.

## Author Contributions

**Conceptualization:** Monde Muyoyeta, Norbert Heinrich, Laura Olbrich, Elizabeth L. Corbett, Nora Engel, Claudia M. Denkinger.

**Data curation:** Yannis Herrmann, Federica Lainati.

**Formal analysis:** Yannis Herrmann, Federica Lainati, María del Mar Castro.

**Funding acquisition:** Claudia M. Denkinger.

**Investigation:** Yannis Herrmann, Federica Lainati, María del Mar Castro, Chanda P. Mwamba, Moses Kumwenda, Nora Engel, Claudia M. Denkinger.

**Methodology:** Yannis Herrmann, Federica Lainati, María del Mar Castro, Shannon A. McMahon, Nora Engel, Claudia M. Denkinger.

**Project administration:** Yannis Herrmann, Chanda P. Mwamba, Moses Kumwenda, Monde Muyoyeta, Nora Engel, Claudia M. Denkinger.

**Resources:** Claudia M. Denkinger.

**Software:** Yannis Herrmann, Federica Lainati.

**Supervision:** Monde Muyoyeta, Nora Engel, Claudia M. Denkinger.

**Validation:** Chanda P. Mwamba, Moses Kumwenda, Tobias Broger, Norbert Heinrich, Elizabeth L. Corbett, Shannon A. McMahon, Nora Engel.

**Visualization:** Yannis Herrmann, María del Mar Castro, Tobias Broger, Nora Engel.

**Writing – original draft:** Yannis Herrmann, Federica Lainati, María del Mar Castro.

**Writing – review & editing:** Chanda P. Mwamba, Moses Kumwenda, Monde Muyoyeta, Tobias Broger, Norbert Heinrich, Laura Olbrich, Elizabeth L. Corbett, Shannon A. McMahon, Nora Engel, Claudia M. Denkinger.

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
