## [Decision Letter · Decision Letter 0]

15 Mar 2022

PGPH-D-21-01184

User perspectives and preferences on a novel TB LAM diagnostic (Fujifilm SILVAMP TB LAM) – a qualitative study in Malawi and Zambia

Dear Dr. Denkinger,

Thank you for submitting your manuscript to PLOS Global Public Health. After careful consideration, we feel that it has merit but does not fully meet PLOS Global Public Health’s publication criteria as it currently stands. Therefore, we invite you to submit a revised version of the manuscript that addresses the points raised during the review process.

We look forward to receiving your revised manuscript.

Kind regards,

Amrita Daftary

Section Editor

Journal Requirements:

Additional Editor Comments (if provided):

Reviewers' comments:

Reviewer's Responses to Questions

**Comments to the Author**

1. Does this manuscript meet PLOS Global Public Health’s publication criteria? Is the manuscript technically sound, and do the data support the conclusions? The manuscript must describe methodologically and ethically rigorous research with conclusions that are appropriately drawn based on the data presented.

Reviewer #1: Partly

Reviewer #2: Partly

2. Has the statistical analysis been performed appropriately and rigorously?

Reviewer #1: No

Reviewer #2: No

3. Have the authors made all data underlying the findings in their manuscript fully available (please refer to the Data Availability Statement at the start of the manuscript PDF file)?

Reviewer #1: Yes

Reviewer #2: No

4. Is the manuscript presented in an intelligible fashion and written in standard English?

Reviewer #1: Yes

Reviewer #2: Yes

5. Review Comments to the Author

Reviewer #1: Dear Author,

Thank you for the opportunity to review this important work. Below I provide feedback on the paper, overall and per section.

Overall feedback on research:

This paper investigates an important opportunity for the timely diagnosis of TB in a high-risk population, utilising a relatively simple process. The potential of FujiLAM to change way PLHIV and children are diagnosed with TB must be brought to the attention of researchers and policy developers alike, particularly in high-burden, resource constrained settings.

Overall feedback on writing:

Check grammar. A few errors were found where authors used the wrong words, or included words that weren't supposed to be in the sentence. E.g., Line 32, should read 'sub-Saharan Africa' not 'Sub-Saharan Africa'.

Aim for clarity in writing. Also, beware of ethics inferred through writing. E.g., Line 11, 'decision makers' is never explained. Who are DMs? What role(s) do they play in the TB programme? Or, the HIV programme? Lines 28, 193, 'respondents' removes agency from people with whom we conduct research, 'participants' is a preferable term. Lines 44-46, 'Abbott’s Alere Determine TB-LAM (USA; henceforth called AlereLAM), has shown to reduce mortality from TB disease in patients with advanced HIV despite a limited sensitivity.' The onus is placed on the reader to determine that the LAM test resulted in earlier TB diagnoses, earlier TB diagnoses resulted in earlier treatment initiation and earlier treatment initiation results in reduced mortality. Line 361, 'patients and children'. Are children not also patients? Similarly in Line 364, 'children and people'. Are children not people?

Feedback on method:

Existing section entitled 'context,' provides very little context to the burden of TB and HIV in Malawi and Zambia. This 'epidemiological' context is important to show the reader why FujiLAM could help change NTPs in similar contexts. In addition to your description of use of LAM testing, it is important to include information about the public health system: do TB patients generally receive care at public hospitals? In other southern African countries, TB patients receive care at primary health care facilities. However, sputum and Xpert tests are not done at these facilities and samples are sent to other health facilities which have the technologies with which to test samples. Without this information it is difficult to determine if, for example, Lines 129-131, are true.

Existing section entitled 'FujiLAM test characteristics and use instructions,' could be summarised elsewhere.

Table 1 could be better presented.

'Data entry and analysis' is confusing. The description of data analysis approach (Lines 156-164) aligns well with File S1. However, then findings are 'presented following the health equity implementation framework' (Line 165). This is not methodologically sound. The description of methods used aligns closely with an inductive thematic analysis approach, whereas the use of the 'equity implementation framework' aligns with a deductive thematic analysis approach. The former allows the data, through iterative analysis, to determine themes. The latter presumes themes included in the equity implementation framework are appropriate to describe the data. Additionally, the sub-themes used in the results section are not introduced nor do they align with File S1 or the 'equity implementation framework.' Consider using the themes arrived at in File S1.

If the authors would like to use the equity implementation framework, then the onus is on them to describe the framework, and illustrate its applicability to the presented study. This is not done. The only mention of the health equity implementation framework is in Line 70, where its appropriateness is presumed.

Feedback on results:

There are 42 interviews, 20 of which are among patients. However, patients views (and quotes) are rarely described in the results. This is strange given the authors argue FujiLAM will improve health equity. Consider being more purposeful in presenting both patients and DMs/HCPs perspectives throughout the results, like you do for the 'recipient factors' theme.

Given the disconnect between the methodological approach and the application of the 'health equity implementation framework,' as well as the lack of explanation of the frameworks' application to the data, the results are difficult to follow. E.g., the existing section, 'Criteria for CD4 count and its potential impact on FujiLAM’s use and implementation' is more about changes in guidelines than the innovation itself. It is unclear how/why it belongs in the 'Innovation factors' theme.

Feedback on discussion:

A strong case is made of the potential of FujiLAM to improve existing health systems practices and policies for NTPs. A good overview of results is provided.

Limitations are adequately noted. However, Lines 540-544, do not adequately acknowledge that 'urban public hospitals' are not the primary facilities in which patients seek TB and HIV care. This should also have been included in the description of the 'context'.

Feedback on conclusion:

This study cannot adequately establish whether FujiLAM can be decentralised to rural clinics, and taken up as part of the diagnostic processes in these clinics, because the data are not from this setting. At most, they can claim HCPs, DMs and patients at two urban public hospitals suggest FujiLAM can and should be decentralised for use in primary health care facilities in Zambia and Malawi.

Reviewer #2: First and foremost,

Congratulations to the Authors for exploring this important subject.

1. The script is generally well written

Minor revisions:

1. There are few spelling errors in the text, e.g. Below in line 96 has been spelled as "bellow", i encourage the authors to check for spelling errors.

Major revisions

1. The methodology contains material relevant in the background, especially the items s stated from line 81 to 117, methodology will flow well if these are moved to the background.

2. In the discussion, the authors have stated the role of the Urine LAM both Fuji and Alere, however, they did not indicate that all LAM positive patients require further evaluation with the Genexpert, especially that WHO classifies Zambia as one of the high MDR TB burden countries, this is useful in ruling or diagnosing resistance to rifampicin.

3. The 1st sentence in the conclusion is not clear, I could not appreciate the what the authors meant by facilitators. Did they mean factors?

4. The reviewers may wish to review the diagnostic algorithm for Zambia, it indicates that LAM can be used in Patients with Sepsis, this is missing in their background statement about the use of LAM in Zambia.

6. PLOS authors have the option to publish the peer review history of their article (what does this mean?). If published, this will include your full peer review and any attached files.

**Do you want your identity to be public for this peer review?** For information about this choice, including consent withdrawal, please see our Privacy Policy.

Reviewer #1: No

Reviewer #2: No

---

## [Editor Report · Decision Letter 1]

30 May 2022

User perspectives and preferences on a novel TB LAM diagnostic (Fujifilm SILVAMP TB LAM) – a qualitative study in Malawi and Zambia

PGPH-D-21-01184R1

Dear Dr. Denkinger,

We are pleased to inform you that your manuscript 'User perspectives and preferences on a novel TB LAM diagnostic (Fujifilm SILVAMP TB LAM) – a qualitative study in Malawi and Zambia' has been provisionally accepted for publication in PLOS Global Public Health.

Best regards,

Amrita Daftary

Section Editor